# Insights into Effects of Combined Capric and Lauric Acid on Rumen Bacterial Composition

**DOI:** 10.3390/microorganisms12061085

**Published:** 2024-05-27

**Authors:** Mariana Vadroňová, Adam Šťovíček, Alena Výborná, Yvona Tyrolová, Denisa Tichá, Miroslav Joch

**Affiliations:** 1Department of Microbiology, Nutrition and Dietetics, Faculty of Agrobiology, Food and Natural Resources, Czech University of Life Sciences, Kamýcká 129, 165 00 Prague, Czech Republic; vadronovam@af.czu.cz (M.V.); stovicek@af.czu.cz (A.Š.); tichad@af.czu.cz (D.T.); 2Department of Nutrition and Feeding of Farm Animals, Institute of Animal Science, Přátelství 815, 104 00 Prague, Czech Republic; vyborna.alena@vuzv.cz (A.V.); tyrolova.yvona@vuzv.cz (Y.T.)

**Keywords:** ruminants, dairy cows, rumen bacteria, capric acid, lauric acid

## Abstract

This study used next-generation sequencing to assess the impact of combined capric acid (C10) and lauric acid (C12) on the ruminal bacterial composition. Eight Holstein cows were randomly assigned to two groups using a cross-over design. The cows were fed two silage-based diets with the addition of either 100 g of stearic acid per cow per day (control), or 50 g of capric acid and 50 g of lauric acid per cow per day (C10 + C12). On day 18, 250 mL of rumen fluid was collected from each cow, and DNA was isolated, amplified, and sequenced. Treatment did not alter bacterial diversity indices, the relative abundance of archaea, nor the fiber-degrading microorganisms, except for a decrease in *Fibrobacter* (from 2.9% to 0.7%; *p* = 0.04). The relative abundance of *Prevotellaceae* decreased (from 39.9% to 29.6%; *p* = 0.009), which is notable because some members help to efficiently utilize ammonia by releasing it slowly into the rumen. Furthermore, the relative abundance of *Clostridia* increased (from 28.4% to 41.5%; *p* = 0.008), which may have aided the increased ammonia–nitrogen levels in the rumen, as this class contains hyperammonia-producing members. Our study reveals alterations in bacterial abundances with implications for rumen ammonia levels, offering insights into potential strategies for modulating rumen fermentation processes and methane production in ruminant livestock.

## 1. Introduction

One of the main targets in the agriculture sector is reducing greenhouse gas (GHG) emissions, specifically those of methane (CH_4_). Of global CH_4_ emissions, 31% originate from agriculture [1]. Most of these emissions originate from enteric fermentation in the bovine rumen, and account for 4.3% of the global GHG emissions [2]. In addition to being a potent GHG, CH_4_ also represents a significant energy loss for the animals (2–12% of gross energy [2]). This energy can be saved and used to increase efficiency and production sustainability [3,4], as the global demand for meat and milk is predicted to increase. To implement sustainable practices in ruminant agriculture, it is essential to understand how anti-methanogenic additives affect the rumen microbiota.

Ruminants have a unique digestive system that heavily relies on symbiotic relationships with their microbiota. Ruminal microorganisms provide the ruminants with up to 70% of their energy requirements in the form of volatile fatty acids (VFAs). Microbial fermentation of feedstuffs also produces carbon dioxide (CO_2_) and hydrogen (H_2_), which are the main substrates for CH_4_ production. Methanogenesis is performed by the methanogenic archaea, which indirectly stabilizes the fermentation process in the rumen by using H_2_ and reducing the H_2_ partial pressure [5]. In addition to archaea, the rumen harbors a highly diverse ecosystem of bacteria, protozoa, and fungi that actively degrade and use different feed components. Disturbances in this complex ecosystem in the form of antimethanogenic additives elicit various reactions [5]. Gaining insight into ruminal microbiota modulation and individual variations in ruminal microbial populations may enhance fermentation, decrease CH_4_ emissions, and potentially help identify and select more efficient animals [6]. To manipulate ruminant efficiency, understanding the role of ruminal microbiota in feed utilization by the host is important.

Numerous studies have modified microbial fermentation and decreased CH_4_ production through changes in diet composition or supplementation with feed additives (e.g., 3-nitrooxypropanol [3-NOP], Asparagopsis). CH_4_-inhibiting feed additives vary in their mechanisms of action. For example, a few target methanogens directly (3-NOP), a few provide a competitive electron acceptor (nitrates), and others increase the production of propionate (probiotics) or exert antimicrobial effects (oils and plant compounds [7]). Among the antimicrobial agents, medium-chain fatty acids (MCFAs) have shown promising inhibitory effects on methanogenesis, with a chain-length-dependent impact. Capric acid (C10) and lauric acid (C12) may be toxic to bacteria, protozoa, and methanogens [8]. This toxicity to microbes, especially to fibrolytic bacteria [9] and protozoa, may affect fiber digestion [5]. Previous studies have either not explored the effects of MCFA on microbiota or used insufficiently specific techniques [10,11,12,13]. The exact effect of the combination of C10 + C12 on ruminal bacteria has not been described and, to the best of our knowledge, has not been fully confirmed. In our previous study, with Joch et al. [14], we described the effects of C10 + C12 on rumen fermentation, gas production, and protozoa counts in vivo. The present study aimed to expand upon these findings and used next-generation sequencing to evaluate the effects of C10 + C12 on ruminal bacteria.

We hypothesized that the supplementation with C10 + C12 would (1) decrease the diversity of the bacterial population due to the predicted antimicrobial effect, (2) decrease the relative abundance of bacteria producing H_2_ (e.g., *Ruminococcaeae*) and increase the relative abundance of H_2_ utilizing bacteria (e.g., *Succinivibrionaceae*), and (3) inhibit fibrolytic bacteria (e.g., *Fibrobacter* spp.).

## 2. Materials and Methods

### 2.1. Ethical Compliance

Animal procedures adhered to Czech legislation (Act No. 246/1992 Coll., on the protection of animals against cruelty) and relevant European directives and regulations (Directive 2010/63/EU, on the protection of animals used for scientific purposes). The cows were housed in an experimental farm located at the Institute of Animal Science (Netluky, Prague, Czech Republic).

### 2.2. Animals, Diets, and Experimental Design

Detailed information about the cows and their feeding was reported in our previous study [14]. Briefly, eight multiparous dry Holstein cows were randomly assigned to two groups with two dietary treatments in a cross-over design and allowed to acclimate for 17 d before rumen sample collection. Dietary treatments were as follows: (1) control (control), silage-based basal diet + 100 g of stearic acid per cow per day (cow/d), (2) capric/lauric acid mixture (C10 + C12), silage-based basal diet + 50 g of capric acid + 50 g of lauric acid/cow/d. Fatty acids were pelleted using wheat bran (100 g of fatty acids and 900 g of wheat bran/kg of pellets as feed). One kilogram of pellets was divided into two equal individual meals and administered daily at 05:00 h and 17:00 h. Each group of four cows was housed in an airflow-controlled chamber.

### 2.3. Rumen Fluid Collection

On day 18 of each experimental period, a representative sample (250 mL) of rumen fluid was collected from each cow 3 h after morning feeding using an oral rumen tube technique [15]. The head of the oral-rumen tube fitted with a strainer was inserted to a depth of approximately 180 cm to reach the central rumen, and the sample was collected in a 500 mL Erlenmeyer flask using a vacuum pump. The first 200 mL of rumen fluid was discarded to minimize saliva contamination. The samples were immediately placed on ice and transported to the laboratory. Subsamples were stored at −80 °C for subsequent microbiome analysis.

### 2.4. DNA Extraction, Polymerase Chain Reaction (PCR), and Amplicon Sequence Variant (ASV) Analysis

DNA extraction was conducted as described previously [16] using a modified method of repeated bead beating and column purification. Briefly, 200 μL of DNA was extracted from each homogenized sample (0.5 mL) of the rumen fluid. A Tecan Infinite M200 spectrometer (Tecan Group, Männedorf, Switzerland) was used to quantify the final DNA yield and quality. The extracted DNA was normalized to the same molecule concentration (5 ng/μL) and stored at −20 °C before the polymerase chain reaction (PCR). The V4 region of the 16S rRNA gene was amplified using the primers 515F (5′-GTGCCAGCMGCCGCGGTAA-3′) and 806R (5′-GGACTACHVGGGTWTCTAAT-3′). The number of PCR cycles was uniform for all samples and maintained as low as possible to prevent the formation of chimeric sequences. The success of the PCR amplification was confirmed through agarose gel (1%) electrophoresis, with the gel stained using SYBR™ Green I Nucleic Acid Gel Stain (Thermo Fisher, Waltham, MA, USA) and then documented with the Gel Doc XR+ System (BioRad, Hercules, CA, USA). All samples were amplified in triplicate.

The PCR products were sequenced using the MiniSeq platform (Illumina, San Diego, CA, USA). Subsequent analysis of the resulting amplicons was performed using the DADA2 pipeline and the SILVA database (v128). The obtained data were normalized to the sample with the lowest sequence depth (30,000 sequences/sample).

### 2.5. Visualization, Assessment, and Statistical Analysis of the Bacterial Community

The relative abundance of bacteria was visualized at various taxonomic levels (> 0.1% of the total sequences) in Python using the “plotly” package. Differences in relative abundances were assessed using the Wilcoxon signed-rank test [17]. Differences were considered statistically significant at *p* < 0.05. Alpha diversity indices (Chao1, Pileou, Shannon, and Simpson) were calculated based on the rarefied amplicon sequence variant (ASV) count table and analyzed using a paired *t*-test [17]. Boxplots illustrating significant differences in relative abundances at the family level were generated using the “ggplot2”, “ggpubr”, and “ggsignif” packages in R software (version 4.3.2). The compositional differences (beta diversity) were assessed and visualized using the “vegan” package in R. The data were analyzed using the metaMDS function through a nonmetric multidimensional scaling (NMDS). Since the stress values of the two-dimensional NMDS were <0.20 (stress = 0.1438446), the ordination patterns were deemed acceptable. The stress values for NMDS were reported after 100 attempts, with the best solution being repeated 30 times.

## 3. Results

Diversity, evenness, and richness were not affected by the treatment, as C10 + C12 did not affect the alpha diversity indices (Simpson, Chao1, Shannon, and Pielou) nor the ASV count (*p* > 0.05; Table 1).

Beta diversity, as visualized using the NMDS (Figure 1), did not show a specific cluster or pattern between the samples. The C10 + C12 samples were located in the upper left quadrant, farther from the control, indicating treatment-dependent differences in bacterial composition. The relative abundances of the control and treatment groups at the phylum, class, family, and genus levels are shown in Figure 2. The highest relative abundance at the phylum level (Figure 2A) was noted for *Bacteroidota* (48.2%) and *Firmicutes* (35.9%) in the control group. The treatment decreased the abundance of *Bacteroidota* to 39.7% and increased that of *Firmicutes* to 48.1%.

The class level (Figure 2B) matched the phylum level, and the relative abundance of *Bacteroidia* (phylum *Bacteroidota*) was higher in the control group (50.9%) than in the treatment group (41.6%). In contrast, the highest relative abundance in the experimental group was that of *Clostridia* (phylum *Firmicutes*; 41.5%), which was higher than that in the control group (28.4%). A decrease in the relative abundance of 2.6% to 0.6% was noted for *Fibrobacteria* in the treatment group.

The significantly different families are shown in Figure 2C, and further details for individual cows are shown in Figure 3. The effect of C10 + C12 was consistent at the family level of microbes between individual animals; in all animals, the treatment significantly decreased and increased the relative abundances of *Prevotellaceae*, *Fibrobacteraceae*, and *Christensenellaceae*, respectively (*p* < 0.05). The relative abundances of *Lachnospiraceae* and *Oscillospiraceae* increased in the treatment group apart from one and two animals, respectively. 

On the genus level (Figure 2D), the relative abundances of *Butyrivibrio*, *Christensenellaceae R-7 group*, *Lachnospiraceae NK3A20 group*, and *Fibrobacter* increased (*p* < 0.05) at the expense of the *Prevotella* genus, which decreased (*p* < 0.05).

## 4. Discussion

In a previous study, we reported the effects of C10 + C12 on gas production and rumen fermentation characteristics [14]. In summary, the supplementation of C10 + C12 in vivo decreased CH_4_ production by 11.5% but increased rumen ammonia-N (NH_3_-N) concentrations (+28.5%) and ammonia gas emissions (+37.2%). The treatment did not affect the total VFA concentration, ruminal pH, or protozoal count. This implies that C10 + C12 has CH_4_ mitigation potential but may compromise the efficiency of N utilization. However, the mechanism of action remains unclear. We hypothesized that the microbial communities would provide insights into these observed effects.

### 4.1. Bacterial Diversity, ASVs, and Indices

Our findings show that the C10 + C12 treatment did not alter bacterial diversity, evenness, or richness (Table 1), contradicting our initial hypothesis. The lack of effects on the overall bacterial indices could be explained by the low doses of MCFA (0.8% DM) [18]. Our results further agree with those of Burdick et al. [8], who used comparatively lower doses of MCFA (0.25% DM; blend of C8, C10, and C12). Ruminal diversity was not affected in our study, implying that MCFA did not jeopardize the overall stability of the ruminal community. In contrast, lower microbial richness, diversity, and bacterial abundance, along with a higher abundance of fiber-degrading bacteria, has previously been associated with higher feed efficiency in dairy cows [19,20]. This is because the resulting metabolic network is simpler and leads to higher concentrations of specific components suitable for the host’s energy requirements [21]. In comparison, in our previous in vitro study evaluating the combined effects of nitrate and various MCFAs, the alpha diversity indices decreased. The differences may be because it was an in vitro study, where we used higher concentrations of MCFA, and added nitrate [16]. Another reason could be using a solvent (ethanol) to dissolve MCFA in the medium in vitro, but not in the in vivo *trial*. This corroborates the theory that saturated fatty acids may require partial dissolution in a buffer or medium to have a significant impact [22].

### 4.2. Microbial Relative Abundances

At the kingdom level, the relative abundances of bacteria and archaea (Table 1, Figure 4) were not altered by the treatment, which is in line with other studies on MCFA in the rumen [8,23]. The lack of an effect on the relative abundance does not align with our previously reported decrease in methanogenesis (−11.5%) from these animals [14]. However, there have been reports of reduced methanogenesis (and methanogen activity) with no change in the methanogen number [24]. Consequently, it has been suggested that the abundance of methanogens may not be the primary factor affecting CH_4_ emissions [3], but rather the composition of the archaeal community and the metabolic activity of each methanogenic species [25]. For example, tea saponin did not affect total methanogen numbers, but significantly decreased methanogenesis (by 8%), which was explained by the reduced gene expression (−76%; [24]); specifically, the expression of the gene encoding methyl-coenzyme M reductase (MCR), an enzyme that catalyzes the last step of CH_4_ production [7]. MCFA may be able to reduce methanogenesis by decreasing the metabolic activity of archaea [26]. Thus, in our study, the reduced methanogenesis may have been caused by the altered composition of the methanogenic community, suppressed metabolism, or both.

The dominant bacterial phyla in both the control and treatment groups were *Bacteroidota* and *Firmicutes* (Figure 2A), which is consistent with most ruminant studies [20,27]. These phyla showed an inverse relationship in their relative abundance, as seen in previous research [21,28]; *Bacteroidota* decreased in the C10 + C12 group, and in parallel, *Firmicutes* increased. This reciprocal adjustment aligns with the sensitivity of *Bacteroidota* to altered ruminal environments (e.g., by the diet), and their propensity to be among the first to react to these alterations [6]. The decrease in *Bacteroidota* resulted in an increased *Firmicutes* to *Bacteroidota* (F:B) ratio. This ratio has been identified as a significant parameter for assessing the microbial impact on the host energy requirements. An increased F:B ratio has been shown to contribute to improved bovine performance, including body weight gain and milk fat yield [29], but there are contradictory reports [30]. A greater abundance of *Firmicutes* indicates a more feed-efficient animal [21,31] with improved structural polysaccharide degradation [32]. Changes in this ratio did not affect digestibility, and we could not assess its effect on production parameters, because the animals in our study were dry cows [14].

We observed a few changes in the relative abundances of bacteria at the family level. The relative abundances of *Christensenellaceae* and *Lachnospiraceae* increased; these families are associated with H_2_ production [33]. In contrast, the bacteria associated with H_2_ utilization remained either unaffected (*Succinivibrionaceae*) or decreased (*Prevotella*) by the treatment [34], which refutes our second hypothesis. The general hypothesis is that CH_4_ emissions depend on the abundance of H_2_-producing and H_2_-utilizing bacteria in the rumen [6,34]. The concentrations of H_2_ are maintained at low levels primarily through methanogenesis, which maintains favorable conditions for continual fermentation and fiber digestion. The inhibition of CH_4_ production leads to elevated H_2_ levels in the rumen [9]. For example, supplementation with a CH_4_ inhibitor, 3-NOP, resulted in a 37-fold increase in H_2_ yield [2]. Notably, research shows that even high H_2_ levels (i.e., 550 μM in comparison with the typical range of 0.2–30 μM) do not compromise fiber breakdown in the rumen [35]. If the accumulated H_2_ is not incorporated into alternative H_2_ sinks, some of the excess H_2_ is simply emitted from the rumen [6,36]. Our results imply there was a surplus of H_2_ in the rumen. We did not measure H_2_ emissions nor dissolved H_2_ in the rumen liquid, but the fiber digestibility was not affected [14]. Thus, our results seem to confirm the hypothesis of Hristov et al. [36] according to which the ruminal microbiota can cope with excess H_2_, and surplus H_2_ was most probably emitted. Possibly, ruminal microbiota may have utilized some of the excess H_2_ for processes such as amino acid and fatty acid synthesis [37]. Microbial biomass fulfills a significant portion protein needs of the host and is thought to be a minor electron sink (0.2–0.3%; [38]). However, we did not measure the microbial protein production in this study.

At the genus level, the relative abundance of the fibrolytic *Fibrobacter* decreased in the treatment groups, whereas that of *Ruminococcus* remained unchanged. This may indicate a higher sensitivity of *Fibrobacter* to C10 + C12, even though it is a Gram-negative bacterium that is reported to be less sensitive to MCFA [11,23]. Gram-negative bacteria are usually thought to be more resilient than Gram-positive bacteria due to the presence of a protective outer membrane [11]. The genera *Fibrobacter* and *Ruminococcus* had low relative abundances in the treatment group (0.7% and 2.43%, respectively); however, even taxa with low-abundance taxa may be necessary for the microbial community [20]. The rumen has a highly adaptable microbial community with overlapping metabolic capabilities [21], which probably supports the unaltered digestibility [14].

In our previous study [14], C10 + C12 increased the concentration of ammonium-N (NH_3_-N) in the rumen fluid and the emission of ammonia (NH_3_) in dairy cows. Our results suggest that these changes may be explained by the effect of C10 + C12 on the relative abundances of bacteria that metabolize nitrogen. As mentioned previously, the relative abundance of the family *Prevotellaceae* was decreased. Members of this family, such as *Prevotella Ruminicola*, are considered to have low deamination activity and release NH_3_ at a lower rate, thus allowing its high utilization efficiency [39]. In contrast, the relative abundance of the class *Clostridia* increased during the treatment. A few members of this class (e.g., *Clostridium sticklandii* and *Clostridium aminophilum*) have high deamination activities and are referred to as hyperammonia-producing bacteria (HAB; [39,40]). HAB can hydrolyze small peptides and deaminate amino acids [39] and have been shown to produce ammonia at rates up to 20 times higher than those of other ammonia-producing bacteria found in the rumen [37]. The higher abundance of this class may have caused a higher production of NH_3_ in the rumen (+28.5%; [14]), which other microorganisms could not utilize. Therefore, the higher production of NH_3_-N in the rumen fluid may have led to lower NH_3_ utilization efficiency and higher NH_3_ emissions (+37.2%) when supplying C10 + C12. Inadequate N utilization in the rumen leads to NH_3_-N excretion and ultimately to nitrous oxide emissions from manure, contributing to air and ground water pollution [39]. Ruminants are responsible for approximately 30% of the total anthropogenic NH_3_ emissions and more than 70% of the NH_3_ emissions from the global livestock, which leads to substantial economic losses and adverse effects on human health [41].

The use of NGS in our study has some limitations. Mainly, NGS does not differentiate between live and dead cells [42,43]. Furthermore, to obtain precise functional characterizations of not only bacteria, but also archaea, more advanced techniques would have to be employed, such as metagenomics and metabolomics. Since we report relative abundances, we cannot be sure of the absolute counts of the microorganisms. Another limitation is the data analysis and data interpretation, since studies on microbiota use different sequencing platforms, different groups of closely related DNA sequences (ASVs or OTUs), different software and packages, and statistical tests.

## 5. Conclusions

Our study highlights the effects of combined capric acid and lauric acid supplementation on rumen microbiota, revealing alterations in bacterial abundances with implications for rumen ammonia levels, offering insights into potential strategies for modulating rumen fermentation processes and methane production in ruminant livestock.

## Figures and Tables

**Figure 1 microorganisms-12-01085-f001:**
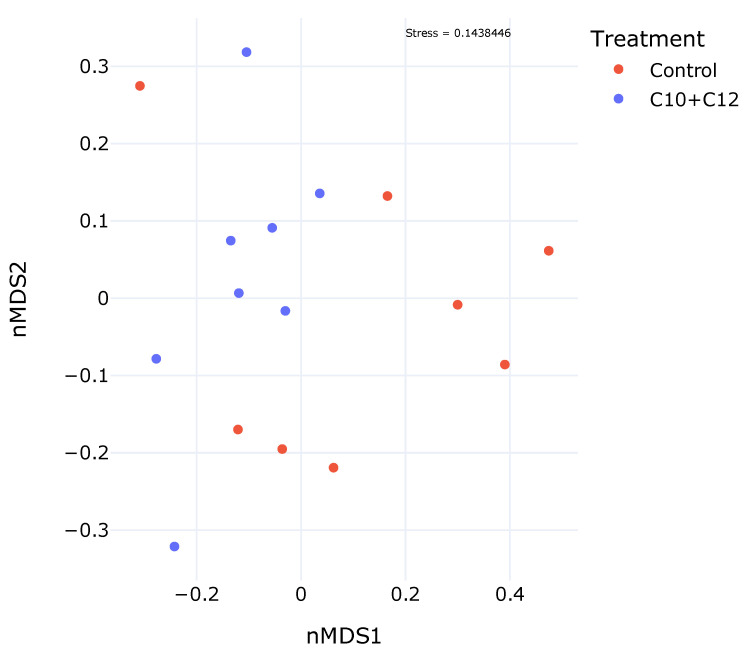
Supplementation with C10 + C12 shows microbiome variations on the amplicon sequence variant (ASV) level through a nonmetric multidimensional scaling (NMDS) visualization based on the control diet (red) or supplemented diet (blue). The C10 + C12 effect is visible in the sample distribution; however, the samples do not aggregate into clusters.

**Figure 2 microorganisms-12-01085-f002:**
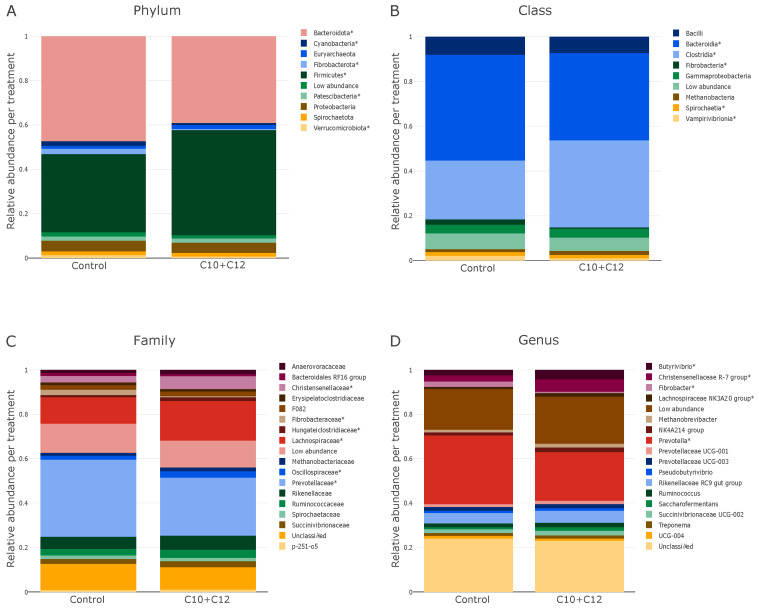
Bar charts showing the effect of C10 + C12 on relative bacterial abundances at the phylum (**A**), class (**B**), family (**C**), and genus (**D**) levels. Taxonomic groups accounting for less than 0.1% of total sequences in each sample were grouped together into a low-abundance category. * Asterisks indicate significant differences at *p* < 0.05.

**Figure 3 microorganisms-12-01085-f003:**
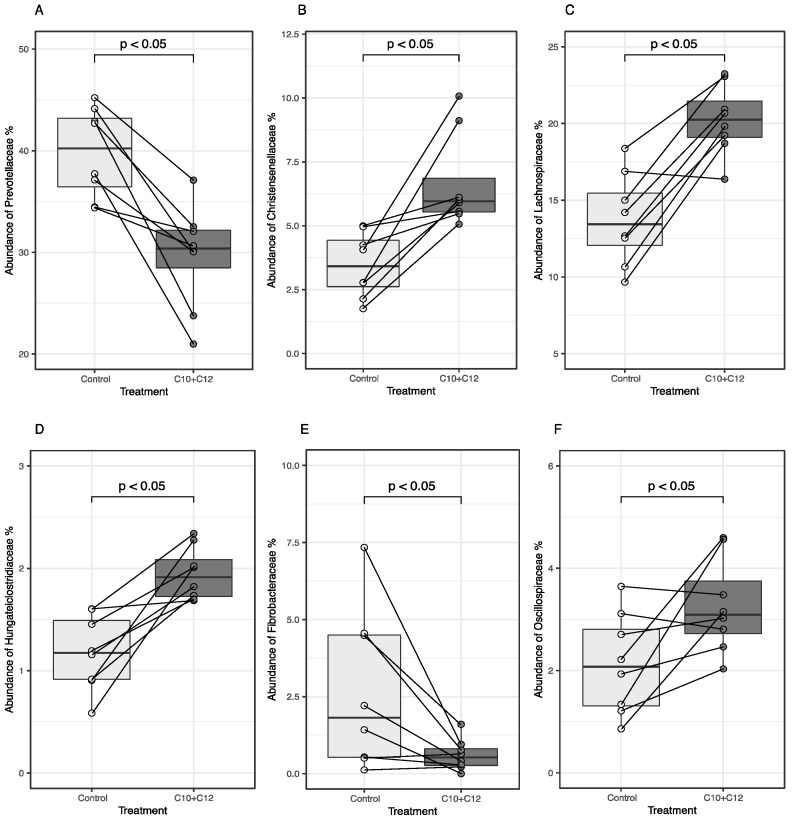
The addition of C10 + C12 to the experimental diet had consistent effects in individual animals on bacterial families (*Prevotellaceae*, (**A**); *Christensenellaceae*, (**B**); *Lachnospiraceae*, (**C**); *Hungateiclostridiaceae*, (**D**); *Fibrobacteraceae*, (**E**); *Oscillospiraceae*, (**F**)). Boxplots show significantly different relative abundances of bacterial families in individual animals without C10 + C12 (control, light gray) supplementation and with supplementation (dark gray) in each animal. The crossover design allowed us to consecutively feed the cows both the control diet and the C10 + C12-supplemented diet (and vice versa). The solid lines connect the paired values of individual cows.

**Figure 4 microorganisms-12-01085-f004:**
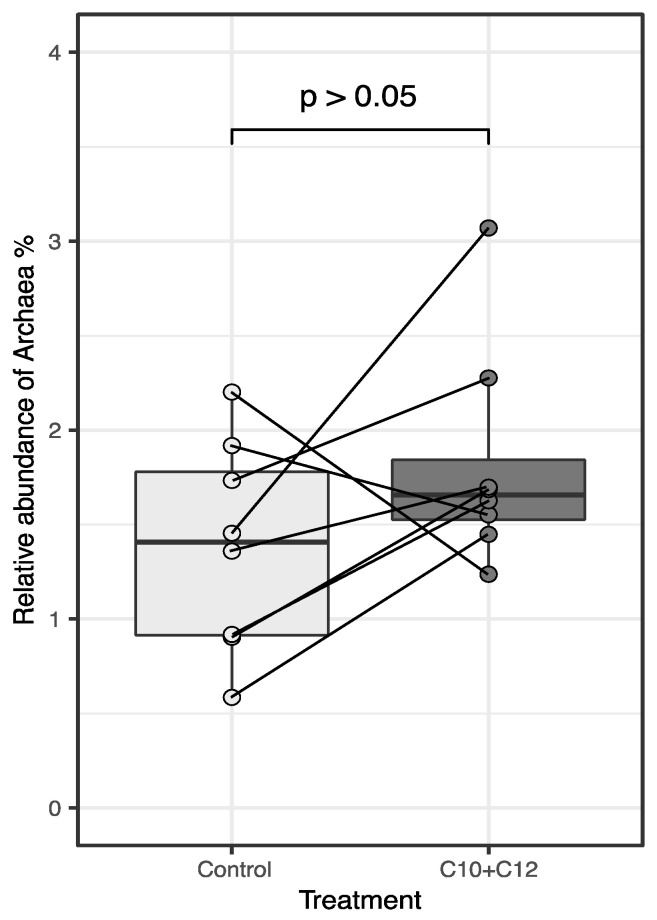
Supplementation of C10 + C12 in the experimental diet did not affect ruminal archaea. The left boxplot (light gray) shows the relative abundances of archaea when the cows were fed the control diet, and the right (dark gray) shows the relative abundances of archaea after supplementation with C10 + C12. The boxplots show a relatively consistent effect on the relative abundance of archaea apart from the two animals. The solid lines connect the paired values of individual cows.

**Table 1 microorganisms-12-01085-t001:** Effect of C10 + C12 on bacterial α-diversity indices, amplicon sequence variant (ASV) count, and methanogen relative abundance in vivo.

Treatment	Shannon Diversity Index	Simpson Diversity Index	Chao1 Richness Index	Pielou Evenness Index	ASV Count	Archaeal Relative Abundance
Control	9.925562	0.995680	5569	0.797338	5589	0.013248
C10 + C12	9.982964	0.996355	5830	0.806313	5334	0.017648
SEM	0.05372	0.0004	22.26162	0.00391	21.45958	0.16
*p*-value	0.2828	0.2918	0.8492	0.1208	0.8956	0.195312

The treatment did not differ significantly (*p* < 0.05) from the corresponding control. SEM, standard error of the mean.

## Data Availability

The raw data supporting the conclusions of this article will be made available by the authors on request.

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
