# Peer review of "Insights into Effects of Combined Capric and Lauric Acid on Rumen Bacterial Composition"

_microorganisms, 2024, doi:10.3390/microorganisms12061085_

Round 1
Reviewer 1 Report
Comments and Suggestions for Authors
The manuscript titled "Insights into Effects of Combined Capric and Lauric Acid on Rumen Microbial Composition" presents a well-structured and comprehensive study on the effects of capric (C10) and lauric acid (C12) supplementation on the rumen microbiota of Holstein cows. The authors employed a cross-over design and next-generation sequencing to elucidate the impact of these medium-chain fatty acids on bacterial diversity and abundance. The study addresses an important issue in agricultural sustainability by exploring potential strategies for methane reduction, which is crucial for mitigating greenhouse gas emissions from livestock.
1. The abstract is concise but could benefit from mentioning key quantitative findings to better inform the reader about the study's impact.
2. Line 26-35: The introduction effectively sets the stage for the study. Consider including more recent references on the global impact of methane emissions and the role of ruminant agriculture in climate change.
3. Hypotheses should be more explicitly stated, providing clear expectations for the study outcomes.
4. Line 88-95: The rumen fluid collection method is well-detailed. Specify the rationale for the 3-hour post-feeding collection time.
5. Line 141-150: The phylum-level findings are significant. Include a brief discussion on the potential implications of changes in Bacteroidota and Firmicutes abundances on animal health and productivity.
6. Line 181-192: The discussion on bacterial diversity aligns with the results. Integrate more references comparing the current findings with other studies using similar treatments.
7. Line 208-239: The microbial relative abundance section is thorough. Address any limitations related to the interpretation of relative abundance data (e.g., potential biases from sample processing).
8. Line 208-239: The microbial relative abundance section is thorough. Address any limitations related to the interpretation of relative abundance data (e.g., potential biases from sample processing).
9. Line 284-289: The conclusion is well-formed. Emphasize the broader implications of the findings for sustainable livestock production.
Author Response
Thank you for taking the time to review, and therefore enhance our manuscript. Please, see the attachment for our point-by-point responses.

Reviewer 2 Report
Comments and Suggestions for Authors
The manuscript Insights into effects of combined capric and lauric acid on rumen microbial composition investigates the effects of supplementing Holstein cows' rations with lauric and capric acid on rumen methanogenesis to reduce methane emissions in ruminants.
The study of mitigation techniques is important to reduce the environmental impact of livestock.
However, the manuscript only reports the results of the bacterial and archaeal community, not the entire microbiota. It is suggested to modify the title accordingly
Abstract
It should be better explained that the work refers to the study of the composition of the ruminal microbiome using next-generation sequencing techniques, otherwise the statement in the first sentence where it says "as this combination previously showed the potential to reduce methane production" is misleading. One wonders what the point of the experiment is. Furthermore, the sentence contradicts the statement in the introduction that there are no other studies on this topic.
Introduction.
Line 29: add reference
Lines 51-55: 3NOP is not part of the experiment, so there is no need to include the sentence, or the concept needs to be expanded with other additives and references.
Line 60: add more references as it says "studies".
Lines 61-62: see above, and it should have been explained here that it is part of a previously published paper.
2.4 are the data available in any database?
2.5 it is reported only bacteria
Line 193: what about archaea? No results on Archeal community are given or discussed except line 209
Line 209. Correct the statement Bacteria is not in fig 4
Figure 2 is difficult to read and the authors did not explain on what basis they selected the reported data as well as the NGS data are available in any database
Fig 2D results are not included
Figure 3. It is not clear from the figure the difference between control and treatment for Fibrobacteriaceae and Oscillospiraceae due to the partial overlap of the box plots.
Use bacterial instead of microbial
Fig 4 only refers to archaea.
No discussion on archaea results
Author Response

(The authors gave the same response as above.)

Reviewer 3 Report
Comments and Suggestions for Authors The present manuscript is a work that provides interesting data regarding the microbiome generated with the addition of capric and lauric acid. However, I do not see in vivo methane measurements, nor do I see that this decrease in methane or modification of the microbiome results in greater efficiency in the use of energy in cows. Or also a better one in milk production. It is an interesting work from the point of view that it provides knowledge, but it falls short for an extensive article. I think that if it is accepted for publication it should be a "Short Communication". The final decision rests with the editor if he decides to make it a long article. It is a quality job but more measurements are missing for a great complete job.Author Response
Thank you for taking the time to read our manuscript. Please, see the attachment for our response.

Round 2
Reviewer 2 Report
Comments and Suggestions for Authors
The manuscript, entitled "Insights into the Effects of Combined Capric and Lauric Acid on Rumen Microbial Composition," examines the impact of supplementing Holstein cows' rations with lauric and capric acid on rumen methanogenesis, to reduce methane emissions in ruminants.
The manuscript has been revised in accordance with the suggestions provided, and the change in the title has resulted in a notable improvement.
A few additional comments have been added to the PFD file.

Author Response
Thank you for your time to evaluate our revision. Please see the attachment for our response.
